# Exposure to low levels of heavy metals and chronic kidney disease in the US population: A cross sectional study

**Akintayo Akinleye** [1] *, **Olayinka Oremade** [2], **Xiaohui Xu** [3]

**1** Department of Internal Medicine, Yale school of Medicine, Yale-Waterbury Internal Medicine Program, Waterbury, Connecticut, United States of America, **2** Department of Patient Safety and Care Improvement, Griffin Hospital, Derby, Connecticut, United States of America, **3** Department of Epidemiology and Biostatistics, Texas A&M University School of Public Health, College Station, Texas, United States of America

* akintayo.akinleye@yale.edu, akintayoakinleye@gmail.com

## Abstract

### Background

Exposure to heavy metals (cadmium, mercury, and lead) has been linked with adverse health outcomes, especially their nephrotoxic effects at high levels of exposure. We conducted a replication study to examine the association of low-level heavy metal exposure and chronic kidney disease (CKD) using a larger NHANES data set compared to previous studies.

### Methods

The large cross-sectional study comprised 5,175 CKD cases out of 55677 participants aged 20–85 years from the 1999–2020 National Health and Nutrition Examination Survey [NHANES]. Logistic regression analysis was applied to estimate the associations between CKD and heavy metals [Cd, Pb, Hg] measured as categorical variables after adjusting with age, race, gender, socioeconomic status, hypertension, diabetes mellitus and blood cotinine level as smoking status.

### Results

Compared to the lowest quartile of blood Cd, exposures to the 2nd, 3rd and 4th quartiles of blood Cd were statistically significantly associated with higher odds of CKD after adjustment for blood Pb and Hg, with OR = 1.79, [95% CI; 1.55–2.07, p<0.0001], OR = 2.17, [95% CI; 1.88–2.51, p<0.0001] and OR = 1.52, [95% CI; 1.30–1.76, p<0.0001] respectively. The 2nd, 3rd and 4th quartiles of blood Cd remained statistically significantly associated with higher odds of CKD after adjustment for blood cotinine level, with OR = 2.06, [95% CI; 1.80–2.36, p<0.0001], OR = 3.18, [95% CI; 2.79–3.63, p<0.0001] and OR = 5.54, [95% CI; 4.82–6.37, p<0.0001] respectively. Exposure to blood Pb was statistically significantly associated with higher odds of CKD in the 2nd, 3rd and 4th quartile groups, after adjustment for all co-variates (ag, gender, race, socio-economic status, hypertension, diabetes mellitus, blood cadmium, mercury, and cotinine levels) in all the four models. Blood Hg level was statistically

**Data Availability Statement:** All relevant data are within the paper and its Supporting information files.

**Funding:** The authors have declared that no competing interests exist.

**Competing interests:** The authors have declared that no competing interests exist.

significantly associated with lower odds of CKD in the 2nd quartile group in model 2, 3rd quartile group in model 1, 2 and 3, and the 4th quartile group in all the four models.

## Conclusions

Our findings showed that low blood levels of Cd and Pb were associated with higher odds of CKD while low blood levels of Hg were associated with lower odds of CKD in the US adult population. However, temporal association cannot be determined as it is a cross sectional study.

## Introduction

Widespread chronic low-dose exposures to heavy metal toxicants such as lead (Pb), mercury (Hg), and cadmium (Cd) have been established through the detection of heavy metals in blood and urine samples of participants in the National Health and Nutrition Examination Survey [1–3]. Heavy metals are well-known environmental pollutants and about a third of the Environmental Protection Agency's National Priority List waste sites are known to contain lead, mercury, or cadmium. Anthropogenic sources of heavy metal pollution include mining, smelting ore, combustion of fossil fuel, and application of phosphate fertilizers [4].

In addition to environmental pollution, other sources of heavy metal toxicants include occupational exposure and consuming foods and products containing heavy metals. For example, exposure to mercury can occur from hazardous spills in the workplace and the consumption of fish from polluted aquatic systems. Cadmium exposure has been associated with cigarette smoking and consumption of vegetables grown with phosphate fertilizers [5], while lead exposure has been linked to lead paint in old homes and the aging water infrastructure [6].

Chronic kidney disease (CKD) has emerged as one of the leading causes of mortality in the United States [7]. CKD is more common in women (14%) than men (12%) and significantly increases the risk of cardiovascular and cerebrovascular diseases. According to the CDC National Chronic Disease Fact Sheet, 15 percent of adults in the United States, equivalent of more than 37 million, have CKD. Patients with CKD that progress to end-stage renal disease eventually require renal replacement therapy with dialysis or a kidney transplant. Of about 100,000 Americans waiting for a kidney transplant in 2020, only 22,817 received one [8].

In addition to chronic diseases like diabetes and hypertension, heavy metal exposure has been suggested to play a role in the development of CKD. Several studies have described the cellular mechanism of nephron injury caused by heavy metal toxins. Mercury has been shown to cause damage to the nucleus and cytoplasm of proximal tubular cells of the nephron [9], cadmium cause a decrease in lactate utilization in proximal tubule cells, impairs glucose production, depletes cellular ATP and acetyl coenzyme A levels and depresses a myriad of enzymatic pathways in the kidney [10] while lead can cause a decline in human mesangial cells in the kidney through oxidative stress damage and programmed cell death [11].

Considering the limited and divergent results of prior studies we conducted a replication study to assess the association of low levels of blood cadmium, lead and mercury with CKD using NHANES data set from 1999–2020. This is a much larger data set than what has been used by previous studies.

## Methods

### Data sources and study population

The NHANES is a cross-sectional, nationally representative survey of the non-institutionalized civilian population of the US and is conducted by the National Center for Health Statistics (NCHS) of the Centers for Disease Control and Prevention (CDC) (NCHS 2008, National Health and Nutritional Examination Survey). The information in the survey contains de-identified data and were fully anonymized by NHANES. The 1999–2020 NHANES dataset was analyzed in this study. 107,622 individuals aged 0 to 80+ years were surveyed for all data cycles.

A total of 58,744 individuals aged 20 years and older were selected for our study. After removing 3,067 individuals with non-positive weights, a total of 55,677 participants remained for final analyses (Fig 1).

### Heavy metal exposure measurements

For the NHANES cycles, whole blood cadmium and lead contents were determined on a PerkinElmer Model SIMAA 6000 simultaneous multi-element atomic absorption spectrometer with Zeeman background correction. Total mercury in whole blood was measured by flow injection cold vapor atomic absorption analysis with online microwave digestion. Serum cotinine was measured by an isotope dilution-high performance liquid chromatography/atmospheric pressure chemical ionization tandem mass spectrometry (ID HPLC-APCI MS/MS). The blood levels of heavy metals were validated by isotope dilution mass spectrometry (IDMS) and inductively coupled plasma mass spectrometry (ICP-MS).

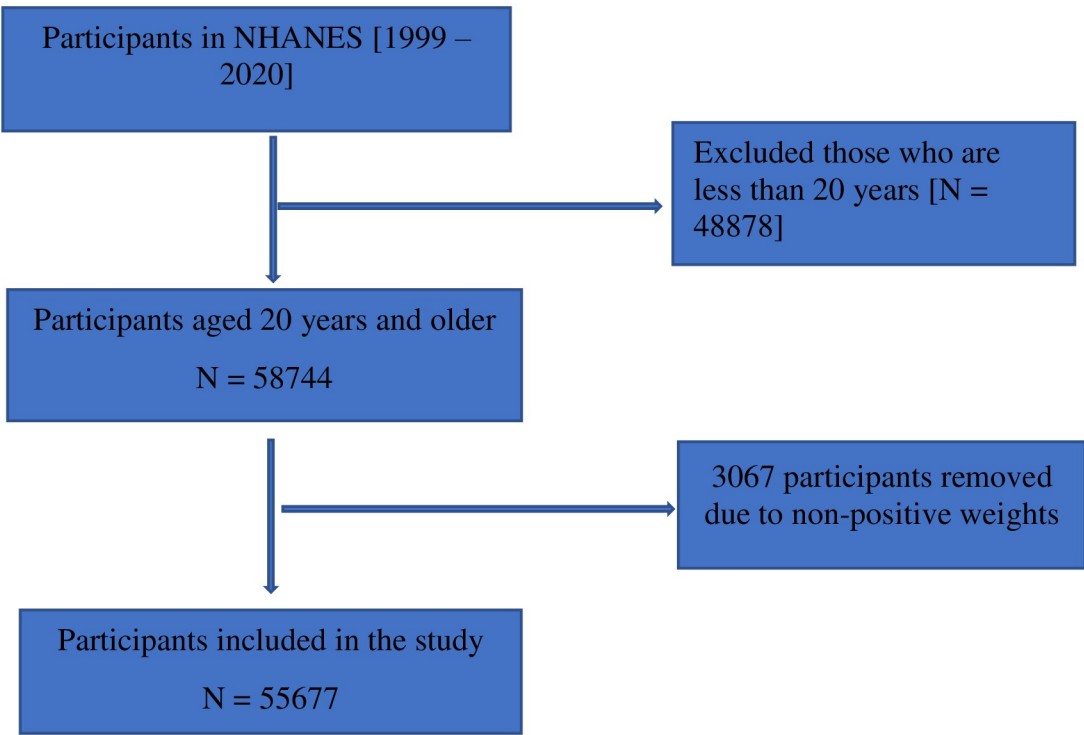

**Fig 1. Flow chart of the sample population in the study.**

**Table 1. The distribution of blood heavy metals in the 1999–2020 NHANES data.**

| Heavy metals | Number of participants | Limits of detection | Range | 25th percentile | 50th percentile | 75th percentile |
|---|---|---|---|---|---|---|
| Blood cadmium (μg/l) | 47572 | 1.10–0.30 | 0.070–13.03 | 0.21 | 0.36 | 0.60 |
| Blood lead (μg/δl) | 47572 | 0.07–0.30 | 0.049–61.29 | 0.90 | 1.30 | 2.10 |
| Blood mercury (μg/l) | 41196 | 0.10–0.33 | 0.070–85.70 | 0.43 | 0.83 | 1.69 |

Method validation was described as "The NHANES quality assurance and quality control (QA/QC) protocols meet the 1988 Clinical Laboratory Improvement Amendments mandates. Detailed QA/QC instructions are discussed in the NHANES LPMs" in the NHANES database.

The blood heavy metal levels were all categorized using quartiles based on the serum levels in the NHANES database.

The smoking history was represented with serum cotinine level. The serum cotinine level was also categorized using quartiles based on the serum levels in the NHANES database.

Each heavy metal has limit of detection documented in different NHANES cycles and can be found in Table 1 which shows the distribution of the heavy metals.

## Ascertainment of CKD cases

Serum creatinine was measured with a modified version of the Jaffe reaction by Popper, Seeling and Wuest. The estimated glomerular filtration rate (eGFR) was calculated using the Modification of Diet in Renal Disease (MDRD) Study formula: GFR (ml/min/1.73m$^2$) = 175 x (S$_{cr}$)$^{-1.154}$ x (Age)$^{-0.203}$ x (0.742 if female) x (1.212 if African American) [12].

The eGFR values less than 60mls/min/1.73m$^2$ were classified as chronic kidney disease based on Kidney Diseases Quality Outcome Initiative (KDOQI) definition [13].

## Covariates

Information about age in years (continuous), gender, race/ethnicity (non-Hispanic White, non-Hispanic Black, Mexican American, Other Hispanic, and Other), and annual household income which was used to calculate the Poverty Index Ratio (PIR) were obtained from the household interview segment of the NHANES data files. PIR less than 1 depicts low socioeconomic status.

History of hypertension was determined if the patient self-reported taking prescription medication for hypertension and history of diabetes mellitus was determined with the patient self-report of "doctor told you, you have diabetes mellitus".

## Analysis

Descriptive analyses, including T-test, ANOVA, and Chi-square test, were used to examine the differences between groups when appropriate. Logistic regression models were applied to test the associations between selected heavy metals and CKD. All analyses were conducted using survey procedures in SAS 9.4 (SAS Institute Inc., College Station, TX, USA), which takes the weighted and clustered sampling design of the NHANES into account.

## Results

Table 2 presents the distribution of selected covariates by CKD status in the 1999–2020 NHANES data.

There were 5,175 participants with CKD, forming 9.3% of the sample population (N = 55,677). The mean age for the study population aged 20–85 years was 47.1 years, with the

**Table 2. Baseline characteristics of the study population (N = 55677) aged (20 to 85 years) in the 1999–2020 National Health and Nutrition Examination Survey (NHANES).**

| Characteristics | CKD cases | | Controls | | Total | |
|---|---|---|---|---|---|---|
| | N | Mean ± SE or Weighted % | N | Mean ± SE or Weighted % | N | Mean ± SE or Weighted % |
| Age [years] | 5175 | 67.69 | 46806 | 45.33 | 55677 | 47.13 ± 0.18 |
| Age in group [years] | | | | | 55677 | |
| 20–39 | 120 | 0.29% | 17544 | 36.82% | 19001 | 37.41% ± 0.45 |
| 40–59 | 723 | 1.80% | 15983 | 35.75% | 17732 | 37.22% ± 0.34 |
| >60 | 4332 | 6.22% | 13279 | 19.11% | 18944 | 25.37% ± 0.41 |
| Gender | | | | | 55677 | |
| Male | 2364 | 3.24% | 22733 | 45.0% | 26793 | 48.03% ± 0.22 |
| Female | 2811 | 5.07% | 24073 | 46.7% | 28884 | 51.97% ± 0.22 |
| Race | | | | | 55677 | |
| Non-Hispanic White | 3222 | 6.75% | 19708 | 61.57% | 24134 | 67.65% ± 1.02 |
| Non-Hispanic Black | 895 | 0.66% | 9881 | 10.0% | 12120 | 11.28% ± 0.56 |
| Mexican American | 460 | 0.25% | 8529 | 7.95% | 9455 | 8.13% ± 0.51 |
| Other race /ethnicity | 292 | 0.37% | 4582 | 6.62% | 5282 | 7.10% ± 0.31 |
| Other Hispanic | 306 | 0.27% | 4106 | 5.55% | 4686 | 5.83% ± 0.43 |
| Hypertension | | | | | 18073 | |
| Yes | 3482 | 17.00% | 12136 | 67.02% | 15618 | 84.01% ± 0.47 |
| No | 136 | 0.72% | 2319 | 15.26% | 2455 | 15.99% ± 0.47 |
| Diabetes Mellitus | | | | | 51979 | |
| Yes | 1485 | 1.97% | 4845 | 7.01% | 6330 | 8.98% ± 0.17 |
| No | 3690 | 6.34% | 41959 | 84.68% | 45649 | 91.02% ± 0.17 |
| Poverty Income Ratio | | | | | 50427 | |
| ≤1 | 846 | 1.00% | 8934 | 13.09% | 10578 | 14.37% ± 0.39 |
| >1 | 3841 | 7.28% | 33652 | 78.63% | 39849 | 85.63% ± 0.39 |
| Heavy metal exposure | | | | | | |
| Mean blood lead level (ug/dl) | 4637 | 2.03 | 42060 | 1.55 | 47572 | 1.59 ± 0.02 |
| Mean blood cadmium level (ug/l) | 4637 | 0.55 | 42060 | 0.51 | 47572 | 0.51 ± 0.01 |
| Mean blood mercury level, total (ug/l) | 3962 | 1.47 | 36492 | 1.57 | 41196 | 1.56 ± 0.03 |
| Mean blood cotinine level (ng/mL) | 5155 | 34.06 | 46688 | 60.78 | 52126 | 58.56 ± 1.43 |
| Outcome variable | | | | | | |
| Estimated Glomerular Filtration Rate, eGFR (ml/min/1.73m$^2$) | 5175 | 48.69 | 46806 | 94.03 | 51981 | 90.26 ± 0.29 |

average age of the group with CKD being 67.7 years and non-CKD participants being 45.3 years. The male to female ratio was 48% to 52%. The study participants were mostly non-Hispanic white race (weighted percentage 67.7%) followed by non-Hispanic black (weighted percentage 11.3%) and the rest were Mexican American, other races and other Hispanic. 10,578 of the study participants (weighted percentage of 14.37%) were of lower socioeconomic status with PIR ≤1. The mean blood lead (Pb) level was 1.59 μg/dl, the mean blood cadmium (Cd) level was 0.51 μg/l, the mean blood mercury (Hg) level was 1.56 μg/l and the mean blood cotinine was 58.56 ng/ml. The mean eGFR for the study population was 90.26 ml/min/1.73m$^2$. There were more people with chronic kidney disease among the age group greater than 60 years with 4,332 out of the total 5,175 people with CKD.

Table 3 shows the crude associations between blood levels of heavy metals (cadmium, lead and mercury) and (CKD).

**Table 3. The crude associations between blood heavy metal levels and chronic kidney disease.**

| Exposure | N | Outcome | | Odds Ratio [OR]/95% [CI] | P-value |
|---|---|---|---|---|---|
| | | CKD | No CKD | | |
| Blood cadmium level(μg/l) | 46697 | | | | |
| Q1 (0.0–0.21) | 11674 | 599 [1.27%] | 11075 [27.72%] | 1.0 | |
| Q2 (0.21–0.36) Vs Q1 | 11487 | 1076 [2.13%] | 10411 [22.79%] | 2.03 [1.78–2.33] | <**0.0001** |
| Q3 (0.36–0.60) Vs Q1 | 10943 | 1384 [2.36%] | 9559 [18.77%] | 2.74 [2.40–3.13] | <**0.0001** |
| Q4 (≥0.60) Vs Q1 | 12593 | 1578 [2.48%] | 11015 [22.47%] | 2.41 [2.11–2.74] | <**0.0001** |
| Blood lead level (μg/δl) | 45713 | | | | |
| Q1 (0.0–0.80) | 11086 | 428 [1.00%] | 10658 [26.44%] | 1.0 | |
| Q2 (0.80–1.30) Vs Q1 | 11675 | 920 [2.03%] | 10755 [24.77%] | 2.17 [1.86–2.55] | <**0.0001** |
| Q3 (1.30–2.10) Vs Q1 | 11846 | 1327 [2.41%] | 10519 [23.13%] | 2.77 [2.38–3.23] | <**0.0001** |
| Q4 (≥2.20) Vs Q1 | 11106 | 1825 [2.76%] | 9281 [17.47%] | 4.19 [3.59–4.90] | <**0.0001** |
| Blood mercury level (μg/l) | 40454 | | | | |
| Q1 (0.0–0.43) | 10076 | 1063 [2.09%] | 9013 [22.37%] | 1.0 | |
| Q2 (0.43–0.83) Vs Q1 | 10053 | 981 [1.96%] | 9072 [22.18%] | 0.95 [0.83–1.08] | 0.40 |
| Q3 (0.83–1.69) Vs Q1 | 10163 | 984 [2.07%] | 9179 [23.30%] | 0.95 [0.83–1.09] | 0.48 |
| Q4 (≥1.69) Vs Q1 | 10162 | 934 [2.07%] | 9228 [23.94%] | 0.92 [0.80–1.07] | 0.29 |
| Blood cotinine level(ng/ml) | 51843 | | | | |
| Q1 (0.0–0.016) | 12896 | 1609 [2.94%] | 11287 [23.87%] | 1.0 | |
| Q2 (0.016–0.05) Vs Q1 | 12938 | 1449 [2.30%] | 11489 [21.70%] | 0.86 [0.78–0.96] | **0.0050** |
| Q3 (0.05–9.81) Vs Q1 | 13056 | 1336 [1.91%] | 11720 [21.49%] | 0.72 [0.65–0.80] | <**0.0001** |
| Q4 (≥9.81) Vs Q1 | 12953 | 761 [1.15%] | 12192 [24.64%] | 0.38 [0.33–0.43] | <**0.0001** |
| Blood cotinine level(ng/ml) | | | | 0.998 [0.997–0.998] | <**0.0001** |

Q1 = <25th percentile, Q2 = 25th to 50th percentile, Q3 = 50th to 75th percentile, Q4 = 75th to 100th percentile

Compared to the lowest quartile of blood Cd, exposures to the 2nd, 3rd and 4th quartiles of blood Cd were statistically significantly associated with higher odds of CKD, with ORs of 2.03 [95% Confidence Interval (CI); 1.78–2.33, p<0.0001], 2.74 [95% CI; 2.40–3.13, p<0.0001] and 2.41 [95% CI; 2.11–2.74, p<0.0001], respectively.

Exposure to blood Pb was statistically significantly associated with higher odds of CKD as compared to the 1st quartile of blood Pb and a dose-response relationship was suggested (OR = 2.17, [95% CI; 1.86–2.55, p<0.0001] in the 2nd quartile; OR = 2.77, [95% CI; 2.38–3.23, p<0.0001] in the 3rd quartile; and OR = 4.19, [95% CI; 3.59–4.90, p<0.0001] in the 4th quartile respectively).

Blood Hg level was not statistically significantly associated with CKD.

Surprisingly, compared to the lowest quartile of blood cotinine level, exposures to the 2nd, 3rd and 4th quartiles of blood cotinine were statistically significantly associated with lower odds of CKD, with ORs of 0.86 [95% CI; 0.78–0.96, p = 0.005], 0.72 [95% CI; 0.65–0.80, p<0.0001] and 0.38 [95% CI; 0.33–0.43, p<0.0001], respectively. This was also supported with the analysis of blood cotinine level as continuous variable with ORs of 0.998, [95% CI; 0.997–0.998, p<0.0001].

Table 4 shows the associations between CKD (eGFR <60 ml/min/1.73m$^2$) and blood heavy metal levels (Cd, Pb, Hg) after adjustment for covariates under four different models. Model 1 was adjusted by the covariates of age, race, gender, poverty index ratio, hypertension and diabetes mellitus; Model 2 was adjusted by the other heavy metals [cadmium, lead and mercury]; Model 3 was adjusted by age, race, gender, poverty index ratio, hypertension, diabetes mellitus and the other heavy metals while Model 4 was only adjusted by blood cotinine level.

**Table 4. Adjusted OR for prevalence of CKD by blood heavy metal levels.**

| Heavy metals | Model 1 | Model 2 | Model 3 | Model 4 |
|---|---|---|---|---|
| | OR [95%CI] | OR [95%CI] | OR [95%CI] | OR [95%CI] |
| Blood cadmium level(μg/l) | | | | |
| Q1 (0.0–0.21) | 1.0 | 1.0 | 1.0 | 1.0 |
| Q2 (0.21–0.36) Vs Q1 | 1.21 [0.98–1.50] | **1.79 [1.55–2.07]** | 1.11 [0.88–1.40] | **2.06 [1.80–2.36]** |
| Q3 (0.36–0.60) Vs Q1 | **1.43 [1.16–1.76]** | **2.17 [1.88–2.51]** | 1.24 [0.98–1.57] | **3.18 [2.79–3.63]** |
| Q4 (≥0.60) Vs Q1 | **1.67 [1.36–2.06]** | **1.52 [1.30–1.76]** | **1.31 [1.05–1.64]** | **5.54 [4.82–6.37]** |
| Blood lead level (μg/δl) | | | | |
| Q1 (0.0–0.80) | 1.0 | 1.0 | 1.0 | 1.0 |
| Q2 (0.80–1.30) Vs Q1 | **1.58 [1.25–2.01]** | **2.12 [1.79–2.50]** | **1.58 [1.23–2.04]** | **2.41 [2.05–2.84]** |
| Q3 (1.30–2.10) Vs Q1 | **1.74 [1.42–2.13]** | **2.75 [2.34–3.22]** | **1.82 [1.46–2.26]** | **3.33 [2.84–3.89]** |
| Q4 (≥2.10) Vs Q1 | **2.39 [1.88–3.03]** | **4.34 [3.66–5.15]** | **2.43 [1.86–3.17]** | **5.65 [4.77–6.67]** |
| Blood mercury level (μg/l) | | | | |
| Q1 (0.0–0.43) | 1.0 | 1.0 | 1.0 | 1.0 |
| Q2 (0.43–0.83) Vs Q1 | 0.91 [0.77–1.07] | **0.87 [0.76–0.99]** | 0.89 [0.74–1.06] | 0.91 [0.80–1.04] |
| Q3 (0.83–1.69) Vs Q1 | **0.82 [0.69–0.97]** | **0.84 [0.73–0.97]** | **0.80 [0.67–0.95]** | 0.89 [0.78–1.02] |
| Q4 (≥1.69) Vs Q1 | **0.80 [0.69–0.95]** | **0.74 [0.64–0.86]** | **0.75 [0.62–0.90]** | **0.84 [0.73–0.97]** |

Model 1: Adjusted for age, race, gender, poverty index ratio, hypertension, diabetes mellitus.

Model 2: Adjusted for other two metals.

Model 3: Model 1+ adjusted for other two metals.

Model 4: Adjusted for blood cotinine level.

## Blood cadmium level and chronic kidney disease

In Model 1, there was a statistically significant association with chronic kidney disease with OR = 1.43, [95% CI; 1.16–1.76, p = 0.0011] in the third and OR = 1.67, [95% CI; 1.36–2.06, p<0.0001] in the fourth quartile respectively.

In Model 2, there was a statistically significant association with chronic kidney disease across all quartiles with OR = 1.79, [95% CI; 1.55–2.07, p<0.0001] in the second, OR = 2.17, [95% CI; 1.88–2.51, p<0.0001] in the third and OR = 1.52, [95% CI; 1.30–1.76, p<0.0001] in the fourth quartile respectively.

In Model 3, there was a statistically significant association between fourth quartile and chronic kidney disease with OR = 1.31, [95% CI; 1.05–1.64, p = 0.0193].

In Model 4 with adjustment for blood cotinine level, all the quartiles of blood cadmium level were statistically significant with chronic kidney disease in a dose response manner with OR = 2.06, [95% CI; 1.80–2.36, p<0.0001] in the second, OR = 3.18, [95% CI; 2.79–3.63, p<0.0001] in the third and OR = 5.54, [95% CI; 4.82–6.37, p<0.0001] in the fourth quartile respectively.

## Blood lead level and chronic kidney disease

All the models showed a statistically significant association in all the quartiles with CKD in a dose response pattern.

In Model 1, there was a statistically significant association with chronic kidney disease across all quartiles with OR = 1.58, [95% CI; 1.25–2.01, p = 0.0002] in the second quartile, OR = 1.74, [95% CI; 1.42–2.13, p<0.0001] in the third and OR = 2.39, [95% CI; 1.88–3.03, p<0.0001] in the fourth quartile respectively.

In Model 2, there was a statistically significant association with chronic kidney disease across all quartiles with OR = 2.12, [95% CI; 1.79–2.50, p<0.0001] in the second, OR = 2.75, [95% CI; 2.34–3.22, p<0.0001] in the third and OR = 4.34, [95% CI; 3.66–5.15, p<0.0001] in the fourth quartile respectively.

In Model 3, there was a statistically significant association with chronic kidney disease across all quartiles with OR = 1.58, [95% CI; 1.23–2.04, p = 0.0005] in the second, OR = 1,82, [95% CI; 1.46–2.26, p<0.0001] in the third and OR = 2.43, [95% CI; 1.86–3.17, p<0.0001] in the fourth quartile respectively.

In Model 4 with adjustment for blood cotinine level, all the quartiles of blood lead level were statistically significant with chronic kidney disease in a dose response manner again with OR = 2.41, [95% CI; 2.05–2.84, p<0.0001] in the second, OR = 3.33, [95% CI; 2.84–3.89, p<0.0001] in the third and OR = 5.65, [95% CI; 4.77–6.67, p<0.0001] in the fourth quartile respectively.

### Blood mercury level and chronic kidney disease

In Model 1, there was a statistically significant association with lower odds of chronic kidney disease with OR = 0.82, [95% CI; 0.69–0.97, p = 0.0217] in the third and OR = 0.80, [95% CI; 0.69–0.95, p = 0.012] in the fourth quartile respectively.

In Model 2, there was a statistically significant association with lower odds of chronic kidney disease across all quartiles with OR = 0.87, [95% CI; 0.76–0.99, p = 0.0468] in the second, OR = 0.84, [95% CI; 0.73–0.97, p = 0.0174] in the third and OR = 0.74, [95% CI; 0.64–0.86, p<0.0001] in the fourth quartile respectively.

In Model 3, there was a statistically significant association with lower odds of chronic kidney disease with OR = 0.80, [95% CI; 0.67–0.95, p = 0.01] in the third and OR = 0.75, [95% CI; 0.62–0.90, p = 0.0018] in the fourth quartile respectively.

In Model 4 with adjustment for blood cotinine level, there was a statistically significant association between fourth quartile and lower odds of chronic kidney disease with OR = 0.84, [95% CI; 0.73–0.97, p = 0.0167].

## Discussion

In this large representative sample of US adults, low-levels of blood cadmium and lead showed a statistically significant association with higher odds of CKD while low-levels of blood mercury were statistically significantly association with lower odds of CKD. Based on the odd ratios, participants in the highest quartile of blood cadmium were more than 5 times more likely to have CKD compared to those in the lowest quartile of blood cadmium. Similarly, participants in the highest quartile of blood lead were more than 5 times more likely to have CKD compared to those in the lowest quartile of blood lead. On the contrary, participants in the highest quartile for blood mercury were 16%–25% less likely to have CKD compared to those in the lowest quartile for blood mercury. For all three metals, the association was consistent in various models.

Our finding of a positive association between low-levels of blood cadmium and CKD was consistent with those by Navas-Acien et al., and Ferraro et al., who both used NHANES 1999–2006 data. It was also consistent with findings from Buser et al., who used NHANES 2007–2012 data and Madrigal et al., who used NHANES 2007–2012 data [14–17]. Similarly, the positive association with low-levels of blood lead was consistent with findings from Navas-Acien et al and Buser et al. but inconsistent with other cross-sectional studies; one that used the Korean National Health and Human Nutrition Survey (KNHANES) and another that used a

combine US and Canada population, both of which did not find a significant association with blood lead and CKD [18, 20].

The negative association between low levels of blood mercury and CKD from our study was inconsistent with a meta-analysis by Jalili et al., [19] and a cross sectional study by Kim et al [18] but consistent with a study in Sweden by Sommar et al. [21] The reason behind the findings is unclear, probably the blood mercury levels found in the US population are not high enough to be nephrotoxic. Further studies will be necessary to explore this finding.

Our study found an inverse relationship between blood cotinine and CKD. Prospective studies by Dulger et al., and Fu et al., have found an independent association between serum cotinine and estimated glomerular filtration rate in the general adult population. The reason behind our finding is unclear at this moment and additional studies will be needed to explore the relationship between cotinine and renal function [22, 23].

A particular strength of our study was its use of the NHANES data which is nationally representative of the US non-institutionalized, civilian population with extensive quality control and high-quality standardized laboratory procedures [24]. Also, the use of 13 cycles of NHANES data in the study with its large sample size increases the power and reduced the random error to the minimum in the association between the heavy metal exposure and the outcome. As at the time of our study, we had the largest sample size exploring all the NHANES data cycles available.

The limitations of our study mirror those of similar studies that used fewer NHANES cycles. Blood heavy metal levels reflect recent exogenous exposure as well as chronic endogenous exposures and as such using blood heavy metal levels as an index of heavy metal exposure remains imperfect. This goes especially for mercury which has a half—life of about 3days to 3 weeks in the body making blood mercury more of an index of acute exposure rather than chronic exposure [25–27]. Another limitation of the study being a cross sectional study is its inability to exclude reverse causation which implies that an increased blood heavy metal level resulted from reduced renal excretion due to kidney damage which result in increased accumulation of heavy metals in the blood. In addition, our study may also have been limited by difficulty in fully adjusting for other potential confounders other than age, gender, race/ethnicity, socioeconomic status, blood cotinine, hypertension, diabetes and other heavy metals. Furthermore, our analysis may also have underestimated the effect of cadmium and lead exposure due to the known limitations of creatinine based eGFR as a marker of chronic kidney disease as compared to eGFR based on serum cystatin C.

In conclusion, although additional prospective studies are needed to fully characterize the association of low levels of blood heavy metals and chronic kidney disease, our findings provide strong support for consideration of low levels of lead and cadmium as CKD risk factors. The data emphasizes the need to continue to monitor and reduce the exposure of cadmium and lead in the general population through the formulation and implementation of policies at all levels of the government agencies to prevent diseases and ultimately safeguard the health of the general population.

## Supporting information

**S1 File.**
(ZIP)

**S2 File.**
(ZIP)

**S3 File.**
(ZIP)

## Author Contributions

**Conceptualization:** Akintayo Akinleye, Olayinka Oremade.

**Formal analysis:** Akintayo Akinleye, Olayinka Oremade.

**Investigation:** Akintayo Akinleye.

**Methodology:** Akintayo Akinleye, Olayinka Oremade, Xiaohui Xu.

**Software:** Akintayo Akinleye.

**Supervision:** Akintayo Akinleye, Xiaohui Xu.

**Writing – original draft:** Akintayo Akinleye, Olayinka Oremade, Xiaohui Xu.

**Writing – review & editing:** Akintayo Akinleye, Olayinka Oremade, Xiaohui Xu.

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
