## [Decision Letter · Decision Letter 0]

25 Jul 2023

PONE-D-23-18872EXPOSURE TO LOW LEVELS OF HEAVY METALS AND CHRONIC KIDNEY DISEASE IN THE US POPULATION: A CROSS SECTIONAL STUDYPLOS ONE

Dear Dr. Akinleye,

Thank you for submitting your manuscript to PLOS ONE. After careful consideration, we feel that it has merit but does not fully meet PLOS ONE’s publication criteria as it currently stands. Therefore, we invite you to submit a revised version of the manuscript that addresses the points raised during the review process. Please submit your revised manuscript by Sep 08 2023 11:59PM. If you will need more time than this to complete your revisions, please reply to this message or contact the journal office at plosone@plos.org. Please include the following items when submitting your revised manuscript:A rebuttal letter that responds to each point raised by the academic editor and reviewer(s). You should upload this letter as a separate file labeled 'Response to Reviewers'.A marked-up copy of your manuscript that highlights changes made to the original version. You should upload this as a separate file labeled 'Revised Manuscript with Track Changes'.An unmarked version of your revised paper without tracked changes. You should upload this as a separate file labeled 'Manuscript'.

We look forward to receiving your revised manuscript.

Kind regards,

Iman Al-Saleh

Academic Editor

PLOS ONE

Journal Requirements:

"NO- The authors have declared that no competing interests exist."

**Additional Editor Comments**:

The authors need to:

1. describe method validation for the analysis of metals and cotinine.

2. present descriptive statistics of analytes in a separate table, including the method detection limit for each metal.

3. discuss the selection of covariates in the model.

4. clarify why only the cadmium model was adjusted for cotinine; there are studies linking other metals with smoking. Have you done a correlation analysis between cotinine and metals?

Reviewers' comments:

Reviewer's Responses to Questions

**Comments to the Author**

1. Is the manuscript technically sound, and do the data support the conclusions?

Reviewer #1: Yes

Reviewer #2: Yes

Reviewer #3: Partly

2. Has the statistical analysis been performed appropriately and rigorously? 

Reviewer #1: Yes

Reviewer #2: Yes

Reviewer #3: Yes

3. Have the authors made all data underlying the findings in their manuscript fully available?

Reviewer #1: Yes

Reviewer #2: Yes

Reviewer #3: Yes

4. Is the manuscript presented in an intelligible fashion and written in standard English?

Reviewer #1: Yes

Reviewer #2: Yes

Reviewer #3: Yes

5. Review Comments to the Author

**Reviewer #1**: The aim of this study was to assess any impact of low blood levels of trace heavy metals and development of CKD. The authors have used a large cohort (NHANES) and included all patients with a diagnosis of CKD, stratified by quartiles of blood Cd, Hg and Pb. The design is fine as a retrospective, descriptive cohort study in that logistic regression can be used to compare odds ratios for the lowest vs highest quartiles of blood heavy metal exposure. The aims and objectives were clearly stated and the outcome clear. I have some comments for the editor and authors to consider:

General comments:

1) I do not really see the relevance of modelling the blood metals as categorical and continuous variables. Surely as categorical has all the information you require?

2) Blood cotinine is surely a serious confounding factor and could affect the logistic regression – in model 3 there appears a significant association between Cd and CKD, but the assoc is hugely strengthened when adjusting for cotinine. Thus presumably this reflects smoking? If smoking AND cotinine are in same model then will have spurious results due to parallel/ co- correlation?

3) Lead accumulates in bone, Cd in kidney>liver and mercury in liver>brain. For the latter it also depends on whether the Hg is organic or inorganic. From fish, mostly organic. To what extent do blood levels reflect bio-accumulation in body tissues or simply acute exposure? ( I recognise you have discussed this as a limitation)

4) If associations are only with 3rd quartile and not 2nd or 4th does that indicate non-linear association with CKD risk or just an anomaly?

5) If all metals show no association after adjustment for all other factors, then do you suggest that the other factors are important (e.g. bioaccumulation with age is the biggest ‘risk’)

6) The final sentence of results and first line of conclusion are therefore at odds.

7) Methods, are the methods for metal determinations published elsewhere? They should be referred to or described.

8) It would be useful to further expand this statement and what you mean by it. Serum cotinine was used to adjust for blood cadmium level.

9) I am also unsure what this sentence actually means, especially the final part…However, blood lead level was only found to be associated with CKD in the 2nd quartile group after adjustment for blood levels of cadmium and mercury but there was a suggested dose-response relationship between the quartile groups prior to adjustment.

**Reviewer #2**: This is a nice paper. However, the most obvious concern lies in the originality of the paper.

Earlier publications using NHANES datasets have addressed the topic. Navas- Acien (2009) is referenced but not Ferraro et al. (2010) Research article Low level exposure to cadmium increases the risk of chronic kidney disease: analysis of the NHANES 1999-2006; Buser et al. (2016) Urinary and blood cadmium and lead and kidney function: NHANES 2007–2012; and Madrigal et al. (2019). Associations between blood cadmium concentration and kidney function in the U.S. population: impact of sex, diabetes and hypertension.

How does this paper compare to these studies. And thus, what additional information does this paper bring to the conversation re. association of low level Cd and CKD risk? The time parameters (1999 -2002) chosen for this subgroup is narrower than other studies (in some, inclusive of the time period); is there a reason for this study window e.g increased exposure risk during that time period?

Suggestion: consider a wider or different time period if there is no supportive justification for the current choice.

There is a very brief recognition of the limitation caused by the exclusion of hypertension - which can cause or arise as a consequence of CKD - but has also been linked to heavy metal exposure. Furthermore, diabetes, another cause of CKD, has also been linked to heavy metal exposure. A lack of recognition of those two chronic diseases as potential confounders are a significant limitation.

Suggestion: Consider including indices of the one/two diseases in at least one of the models to improve paper.

**Reviewer #3**: This is a report in which NHANES data was analyzed to determine if there are correlations between heavy metal exposure and CKD. The analysis seems to indicate that blood levels of Cd were associated with CKD. The major concern associated with this conclusion is that CKD itself could cause high blood levels of heavy metals or other toxicants due to lack of clearance. This concept is mentioned in the discussion, but the abstract needs to be updated to reflect this possibility.

1. Heavy metals are “toxicants” not “toxins.” A toxin is a biological substance (poison/venom) whereas a toxicant is something harmful in the environment.

2. Methods, 1st paragraph, last line: “individuals with missing weighing…” Should this be “weight?”

3. Cadmium and lead should not be capitalized within a sentence.

4. Was there a relationship between blood cotinine levels and CKD stage? Smoking is a major risk factor for CKD and thus, it may be more of factor in the development of CKD than Cd.

5. There are no page number or line numbers so it is difficult to provide feedback on specific grammatical errors. The discussion contains several errors that should be corrected. For example, “Chronic Renal Disease” should be chronic kidney disease.

6. It is not clear which groups are being compared for statistical significance in the tables. Is each quartile being compared to Q1 or is it compared to the previous quartile? This should be clarified in a legend associated with the table.

6. PLOS authors have the option to publish the peer review history of their article (what does this mean?). If published, this will include your full peer review and any attached files.

Reviewer #1: No

Reviewer #2: No

Reviewer #3: No

---

## [Author Response · Author response to Decision Letter 0]

5 Dec 2023

1. describe method validation for the analysis of metals and cotinine.

Response: The method validation has been included in the method section. It was described as “The NHANES quality assurance and quality control (QA/QC) protocols meet the 1988 Clinical Laboratory Improvement Amendments mandates. Detailed QA/QC instructions are discussed in the NHANES LPMs”.

2. present descriptive statistics of analytes in a separate table, including the method detection limit for each metal.

Response: This has been added as table 1 in the method section.

3. discuss the selection of covariates in the model.

Response: The covariates included in the models are listed under table 3. The covariates were identified based on the literature. We have tested the robustness of our results using different models with different sets of the covariates.

4. clarify why only the cadmium model was adjusted for cotinine; there are studies linking other metals with smoking. Have you done a correlation analysis between cotinine and metals?

Response: We have updated model 4 by the adjustment of each heavy metal for blood cotinine level. Previously, only blood cadmium level was adjusted for blood cotinine level.

Reviewer #1: 

General comments:

1) I do not really see the relevance of modelling the blood metals as categorical and continuous variables. Surely as categorical has all the information you require?

Response: Thank you for the suggestion. The continuous variable in the modelling has been removed.

2) Blood cotinine is surely a serious confounding factor and could affect the logistic regression – in model 3 there appears a significant association between Cd and CKD, but the assoc is hugely strengthened when adjusting for cotinine. Thus presumably this reflects smoking? If smoking AND cotinine are in same model, then will have spurious results due to parallel/ co- correlation?

Response: The blood cotinine level was used as a biomarker for tobacco use exposure. We used blood cotinine level to represent smoking status in our analysis. We did not adjust for both blood cotinine level and smoking history which would be duplicative. Upon increasing the sample size in the analysis, significant association between Cd and CKD is found in multiple models. Each heavy metal was adjusted for blood cotinine level in model 4.

3) Lead accumulates in bone, Cd in kidney>liver and mercury in liver>brain. For the latter it also depends on whether the Hg is organic or inorganic. From fish, mostly organic. To what extent do blood levels reflect bio-accumulation in body tissues or simply acute exposure? ( I recognise you have discussed this as a limitation)

Response: Thank you for highlighting this important issue. However, it is difficult to determine if blood levels only reflect acute exogenous or chronic endogenous. This was mentioned in the discussion section. Further prospective studies would be required to answer this question.

4) If associations are only with 3rd quartile and not 2nd or 4th does that indicate non-linear association with CKD risk or just an anomaly?

Response: We have expanded the data analysis to include all the data cycles on NHANES from 1999 to 2020. The positive associations were now found in all the quartiles in all the models.

5) If all metals show no association after adjustment for all other factors, then do you suggest that the other factors are important (e.g. bioaccumulation with age is the biggest ‘risk’)

Response: We think the results might be explained in multiple ways. A small sample size would be one of the possible explanations. Upon increasing the sample size, we found a positive significant association between blood cadmium and lead with chronic kidney disease. We updated the results in the revised manuscript. 

6) The final sentence of results and first line of conclusion are therefore at odds.

Response: This has been rectified. We concluded that there is an association between low levels of cadmium and lead and chronic kidney disease.

7) Methods, are the methods for metal determinations published elsewhere? They should be referred to or described.

Response: The methods for metal determination have been included in the method section.

8) It would be useful to further expand this statement and what you mean by it. Serum cotinine was used to adjust for blood cadmium level.

Response: Thank you for this comment again. We updated our analyses in model 4, which accounted for serum cotinine as an additional covariate when we studied for the effects of each metal. We revised the sentence to reflect this analysis to say “The smoking history was represented with serum cotinine level. The serum cotinine level was also categorized using quartiles based on the serum levels in the NHANES database”.

9) I am also unsure what this sentence actually means, especially the final part…However, blood lead level was only found to be associated with CKD in the 2nd quartile group after adjustment for blood levels of cadmium and mercury but there was a suggested dose-response relationship between the quartile groups prior to adjustment.

Response: This has been rectified in the result section. The result section has been revised entirely as we increased the sample size in the latest analysis.

Reviewer #2: This is a nice paper. However, the most obvious concern lies in the originality of the paper.

Earlier publications using NHANES datasets have addressed the topic. Navas- Acien (2009) is referenced but not Ferraro et al. (2010) Research article Low level exposure to cadmium increases the risk of chronic kidney disease: analysis of the NHANES 1999-2006; Buser et al. (2016) Urinary and blood cadmium and lead and kidney function: NHANES 2007–2012; and Madrigal et al. (2019). Associations between blood cadmium concentration and kidney function in the U.S. population: impact of sex, diabetes and hypertension.

How does this paper compare to these studies. And thus, what additional information does this paper bring to the conversation re. association of low level Cd and CKD risk? 

Response: We have increased the sample size to include additional recent NHANES data cycles from 1999-2020 exploring the associations of low levels of cadmium, lead and mercury and risk of CKD. We have also used different models to test the robustness of our results by adjusting for different sets of covariates in our analyses. The potential dose-response relationships between the metals and CKD offered additional insights about the causal relationships. 

The time parameters (1999 -2002) chosen for this subgroup is narrower than other studies (in some, inclusive of the time period); is there a reason for this study window e.g increased exposure risk during that time period?

Suggestion: consider a wider or different time period if there is no supportive justification for the current choice.

Response: We have increased the sample size to capture 1999 to 2020.

There is a very brief recognition of the limitation caused by the exclusion of hypertension - which can cause or arise as a consequence of CKD - but has also been linked to heavy metal exposure. Furthermore, diabetes, another cause of CKD, has also been linked to heavy metal exposure. A lack of recognition of those two chronic diseases as potential confounders are a significant limitation.

Suggestion: Consider including indices of the one/two diseases in at least one of the models to improve paper.

Response: Upon reviewer’s suggestion, hypertension and diabetes mellitus have been included as confounders in two of the models. The results were presented in table 4.

Reviewer #3: This is a report in which NHANES data was analyzed to determine if there are correlations between heavy metal exposure and CKD. The analysis seems to indicate that blood levels of Cd were associated with CKD. The major concern associated with this conclusion is that CKD itself could cause high blood levels of heavy metals or other toxicants due to lack of clearance. This concept is mentioned in the discussion, but the abstract needs to be updated to reflect this possibility.

Response: This has been updated in the abstract. Additional comment “However, temporal association cannot be determined as it is a cross sectional study” to reflect this.

1. Heavy metals are “toxicants” not “toxins.” A toxin is a biological substance (poison/venom) whereas a toxicsant is something harmful in the environment.

Response: Thank you for the suggestion. We have made all changes in the manuscript. 

2. Methods, 1st paragraph, last line: “individuals with missing weighing…” Should this be “weight?”

Response: Thank you, reviewer. This has been changed to “non-positive weights” in the method section and figure 1.

3. Cadmium and lead should not be capitalized within a sentence.

Response: Thank you for the suggestion. This has been corrected in the manuscript.

4. Was there a relationship between blood cotinine levels and CKD stage? Smoking is a major risk factor for CKD and thus, it may be more of factor in the development of CKD than Cd.

Response: The association between heavy metals and CKD was adjusted by blood cotinine level in model 4. The result still showed association between blood Cd and Pb and CKD after adjustment for blood cotinine.

5. There are no page number or line numbers so it is difficult to provide feedback on specific grammatical errors. The discussion contains several errors that should be corrected. For example, “Chronic Renal Disease” should be chronic kidney disease.

Response: Thank you for the suggestion. Page number has been included and the article has been reviewed for grammatical errors.

6. It is not clear which groups are being compared for statistical significance in the tables. Is each quartile being compared to Q1 or is it compared to the previous quartile? This should be clarified in a legend associated with the table.

Response: It has been clarified to use Q1 as the reference level in both tables 3 and 4.

---

## [Decision Letter · Decision Letter 1]

27 Dec 2023

PONE-D-23-18872R1EXPOSURE TO LOW LEVELS OF HEAVY METALS AND CHRONIC KIDNEY DISEASE IN THE US POPULATION: A CROSS SECTIONAL STUDYPLOS ONE

Dear Dr. Akinleye, Thank you for submitting your manuscript to PLOS ONE. After careful consideration, we feel that it has merit but does not fully meet PLOS ONE’s publication criteria as it currently stands. Therefore, we invite you to submit a revised version of the manuscript that addresses the points raised during the review process.

We look forward to receiving your revised manuscript.

Kind regards,

Iman Al-Saleh

Academic Editor

PLOS ONE

Journal Requirements:

**Additional Editor Comments:**

**Reviewer #2** has proposed two minor additions outlined below. I would appreciate it if you could integrate these suggestions.

Reviewers' comments:

Reviewer's Responses to Questions

**Comments to the Author**

1. If the authors have adequately addressed your comments raised in a previous round of review and you feel that this manuscript is now acceptable for publication, you may indicate that here to bypass the “Comments to the Author” section, enter your conflict of interest statement in the “Confidential to Editor” section, and submit your "Accept" recommendation.

Reviewer #2: All comments have been addressed

Reviewer #3: All comments have been addressed

2. Is the manuscript technically sound, and do the data support the conclusions?

Reviewer #2: Yes

Reviewer #3: Yes

3. Has the statistical analysis been performed appropriately and rigorously? 

Reviewer #2: Yes

Reviewer #3: Yes

4. Have the authors made all data underlying the findings in their manuscript fully available?

Reviewer #2: Yes

Reviewer #3: Yes

5. Is the manuscript presented in an intelligible fashion and written in standard English?

Reviewer #2: Yes

Reviewer #3: Yes

6. Review Comments to the Author

**Reviewer #2: 1. What are the possible explanations for the findings with cotinine?**

**2. Provide the reference for the classification of CKD. (KDIGO guidelines/KDOQI?)**

Reviewer #3: The authors have addressed all of the previous concerns. This reviewer has no additional concerns .

7. PLOS authors have the option to publish the peer review history of their article (what does this mean?). If published, this will include your full peer review and any attached files.

Reviewer #2: No

Reviewer #3: No

---

## [Author Response · Author response to Decision Letter 1]

7 Feb 2024

Dear Editor,

Thank you and the reviewers for the constructive suggestions and reviews. We have made changes in the revised manuscript to reflect the editor’s and reviewers’ comments.

Reviewer #2: 

1. What are the possible explanations for the findings with cotinine?

Thank you for the suggestion. The reason behind the inverse relationship between blood cotinine level and CKD is unclear at this moment. Different studies have found contrasting relationships between them. Further studies would need to be done to explore the relationship.

2. Provide the reference for the classification of CKD. (KDIGO guidelines/KDOQI?)

Thank you for the suggestion. The CKD classification has been referenced with k/DOQI.

---

## [Editor Report · Decision Letter 2]

12 Feb 2024

EXPOSURE TO LOW LEVELS OF HEAVY METALS AND CHRONIC KIDNEY DISEASE IN THE US POPULATION: A CROSS SECTIONAL STUDY

PONE-D-23-18872R2

Dear Dr. Akinleye,

We’re pleased to inform you that your manuscript has been judged scientifically suitable for publication and will be formally accepted for publication once it meets all outstanding technical requirements.

Kind regards,

Iman Al-Saleh

Academic Editor

PLOS ONE
---

## [Editor Report · Acceptance letter]

1 Apr 2024

PONE-D-23-18872R2 

PLOS ONE

Dear Dr. Akinleye, 

I'm pleased to inform you that your manuscript has been deemed suitable for publication in PLOS ONE. Congratulations! Your manuscript is now being handed over to our production team.

Kind regards, 

on behalf of

Dr. Iman Al-Saleh 

Academic Editor

PLOS ONE